



# Numerical Investigation of Aerodynamic Performance of Wind Turbine Airfoils with Ice Accretion

Khaled Yassin[1], Hassan Kassem[2], Bernhard Stoevesandt[2], Thomas Klemme[3], and Joachim Peinke[1]

[1]ForWind-Centre for Wind Energy Research, Institute of Physics, University of Oldenburg, 26129, Oldenburg, Germany
[2]Fraunhofer Institute for Wind Energy Systems, Küpkersweg 70, 26129 Oldenburg, Germany
[3]Senvion GmbH, Überseering 10, 22297 Hamburg, Germany.

**Correspondence:** Khaled Yassin (khaled.yassin@uni-oldenburg.de)

**Abstract.** One of the emerging problems in modern computational fluid dynamics is the simulation of flow over rough surfaces, wind turbine blades with ice on its surface for instance. An alternative method to numerically simulate rough surfaces instead of using a grid with y+ < 1 criterion is to use rough wall functions (RWFs) that models the flow behavior in case of the presence of roughness. This work aims to investigate different rough wall function models to find out the model that can provide the most accurate results with the lowest computational resources possible. This aim was achieved by comparing coefficients of lift and pressure resulting from CFD simulations with wind tunnel results of an airfoil with actual ice profiles collected from the site. After implementing new RWFs in OpenFOAM framework and validating the results with published experimental results, the comparison showed that momentum RWF provided the best agreement between simulation and experimental results while using only 25% of the number of cells used with smooth RWF. The conclusions of this work will be applied in a simulation code within OpenFOAM framework to simulate airflow fields of wind turbines with ice accretion.

## 1 Introduction

For the last few decades, energy generation from wind has increased rapidly as a renewable source of energy. Because of this rapidly increasing demand for wind energy, wind turbine manufacturers have been trying to increase the size of the turbine rotors. By increasing rotor diameter and tower height, the single turbine can now harvest more power from wind in almost the same horizontal area occupied by old, small size wind turbines. However, some of the best wind energy sites like in Europe and North America suffer from icing atmospheric conditions. These conditions can decrease the annual energy production by up to 40% as shown by Sailor et al. (2008). This drop in blades' aerodynamic performance occurs due to several detachments and reattachment areas on the surface due to the presence of rough ice surface.

Since the 1930s, different experiments were conducted in order to investigate the effect of roughness on fluid flow in general. One of the pioneers in this filed was Nikuradse (1933) who investigated turbulent flow in pipes with different relative roughness values and Reynolds numbers between $10^4$ to $10^6$. He noticed that the boundary layer follows the log-law as in the case of





smooth surface. However, in case of rough surface, there is a clear velocity shift ($\Delta u$). A few years later, Schlichting (1937) studied the internal flow of a square-section channel with one rough wall. This rough wall had spherical segments, cones

and angular shapes. He found that the velocity shift depends on four different geometrical parameters of roughness elements, namely: element's height, the area projected on the surface, area projected in flow direction and the average distance between elements. Accordingly, he derived an equation to calculate the friction coefficient on the plate surface. Later, Moody (1944) provided an engineering method to calculate friction losses in pipes due to roughness.

For external flow applications, Blanchard (1977) and Hosni et al. (1991) provided measurements of heat transfer performance

of a heated flat plate with hemispherical roughness elements. Also, Hosni et al. (1993) studied the effect of roughness shape on the boundary layer. Achenbach (1977) studied heat transfer of a roughened circular cylinder at different roughness heights and Reynolds numbers.

Form computational point of view, it is known that to correctly model the viscous sub-layer in computational fluid dynamics (CFD) methods, the height of the grid's first cell should fulfill the $y^+ \leq 1$ criteria. Additionally, to avoid high aspect ration cells,

large number of cells over the surface should be generated. Accordingly, one method to simulate rough surfaces is to generate a fine grid that fits the rough surface. Wang and Zhu (2018) simulated ice accretion on the NREL Phase VI wind turbine blade. In their numerical setup, the used a grid $y^+ \approx 1$, which resulted in a grid with total of 7.3 million cells for a half-cylindrical domain for one blade. This gives a good idea about how much computational cost is needed to perform such simulations in the standard way. However, this cost can be much more increased for larger turbine blades manufactured nowadays.

Another approach is to modify the mathematical models in a way that the behaviour of flow over rough surfaces is grasped without the need for a fine grid. The growing use of CFD in fluid flow simulations in both research and industrial applications has caused an increased interest in this approach and more new models have been developed. The challenge is to find the proper mathematical models which are often based on the results of the aforementioned experiments. Chen and Patel (1988) introduced a two-layer model and a wall function in the k-$\epsilon$ model to simulate rough surfaces. Hellsten and Laine (1997)

provided an extension for the k-$\omega$-SST turbulence model to simulate the behaviour of flow, so did Wilcox et al. (2006) with the k-$\omega$ model. Instead of providing changes to turbulence models, rough wall functions (RWFs) were developed to simulate the behaviour of turbulent flow near walls. In general, wall functions must provide adequate handling for flow change from its stream velocity to stagnation on the wall according to no-slip condition. Thus RWFs should provide additional models that account for roughness. Recently, Da Silva et al. (2011) applied a new $\nu_t$ wall function base on the work of Suga et al. (2006).

Also, Knopp et al. (2009) and Chedevergne and Aupoix (2017) provided k and $\omega$ wall functions that are capable to adapt for the presence of roughness on the surface.

In our work here, we aim to find a computational model that is able to simulate the airflow around iced wind turbine airfoil. This model should provide the most accurate results using the least computational resources possible. This aim is approached by comparing the CFD simulation results of different RWFs with experimental results of actual wind turbine ice accretion

profiles. The studied ice profiles were collected from the site and tested in a wind tunnel after being molded and attached to the airfoil. In this work, three different RWFs were implemented in OpenFOAM® v6 (Greenshields (2017)) and compared: Knopp et al. (2009) RWF , which will be referred in this work as DLR-RWF, Da Silva et al. (2011) RWF, which will be referred as



Momentum-RWF, and Chedevergne and Aupoix (2017) RWF fitted to Colebrook's experimental results, which will be referred as Colebrook-RWF.

This article is structured as follows, Section 2 explains the mathematical model of each of the used RWFs. After that, Section 3 explains the preparation of the rough ice for the simulation and the corresponding computational mesh generation. Section 4 provides the results of each case studied in this work. The last section provides discussion and conclusions of this work and how these results can be applied to general rough surfaces simulations.

## 2 Mathematical Models

In this section, a brief description of the RANS models used in flow field simulations are introduced as a background for the studied RWFs. Subsequently, the details of the implemented RWFs are also explained given.

In RANS models, Navier-Stokes equations are solved as time-averaged while extra one equation is solved (like in Splart-Allmaras) or two equations (like in k-$\omega$ SST) or more equations are solved to close the system of equations required to solve the flow field (Versteeg and Malalasekera (2007)).

### 2.1 RANS turbulence models

In this work, only steady-state RANS turbulence models, namely Spalart-Allmaras and Menter k-$\omega$ SST, were used to simulate the turbulent flow field over the iced airfoils.

#### 2.1.1 Splart-Allmaras Turbulence Model

Splart-Allmaras Turbulence Model is a one-equation model that solves the viscousity-like variable ($\tilde{\nu}$) using the following
equations from Spalart and Allmaras (1992):

$$\frac{\partial \tilde{\nu}}{\partial t} + u_j \frac{\partial \tilde{\nu}}{\partial x_j} = C_{b1}(1 - f_{t2})\tilde{S}\tilde{\nu} - \left[C_{w1}f_w - \frac{C_{b1}}{\kappa^2}f_{t2}\right]\left(\frac{\tilde{\nu}}{d}\right)^2 + \frac{1}{\sigma}\left[\frac{\partial}{\partial x_j}((\nu + \tilde{\nu})\frac{\partial \tilde{\nu}}{\partial x_j}) + C_{b2}\frac{\partial \tilde{\nu}}{\partial x_i}\frac{\partial \tilde{\nu}}{\partial x_i}\right] \tag{1}$$

where

$$\mu_t = \rho\tilde{\nu}f_{v1}, f_{v1} = \frac{\chi^3}{\chi^3 - C_{v1}^3}, \chi = \frac{\tilde{\nu}}{\nu}, \tilde{S} = \Omega + \frac{\tilde{\nu}}{\kappa^2 d^2}f_{v2}$$

#### 2.1.2 k-$\omega$ SST Turbulence Model

k-$\omega$ SST Turbulence Model is a two-equation model that can calculate both turbulence kinetic energy (k) and turbulent frequency ($\omega$) using the equations from Menter (1994):

$$\frac{\partial \rho k}{\partial t} + \frac{\partial \rho u_j k}{\partial x_j} = P - \beta^* \rho \omega k + \frac{\partial}{\partial x_j}[(\mu + \sigma_k \mu_t)\frac{\partial k}{\partial x_j}] \tag{2}$$

$$\frac{\partial \rho \omega}{\partial t} + \frac{\partial \rho u_j \omega}{\partial x_j} = \frac{\gamma}{\nu_t}P - \beta \rho \omega^2 + \frac{\partial}{\partial x_j}[(\mu + \sigma_\omega \mu_t)\frac{\partial \omega}{\partial x_j}] + 2(1 - F_1)\frac{\rho \sigma_{\omega 2}}{\omega}\frac{\partial k}{\partial x_j}\frac{\partial \omega}{\partial x_j} \tag{3}$$



WIND
ENERGY
SCIENCE
DISCUSSIONS
european academy of wind energy

where

$$P = \tau_{ij}\frac{\partial u_i}{\partial x_j}, \tau_{ij} = \mu_t(2S_{ij} - \frac{2}{3}\frac{\partial u_k}{\partial x_k}, \delta_{ij}) - \frac{2}{3}\rho k\delta_{ij}, S_{ij} = \frac{1}{2}\left(\frac{\partial u_i}{\partial x_j} + \frac{\partial u_j}{\partial x_i}\right) \mu_t = \frac{\rho a_1 k}{max(a_1\omega,\Omega F_2)}$$

## 2.2 Rough wall functions

Nikuradse (1933) found that the boundary layer log-law behavior in flow over both rough and smooth surfaces are the same except for a velocity shift ($\Delta u$). The difference between most of the proposed RWFs in various literature is how to calculate

this shift. Schlichting (1937) worked out the parameters on which the velocity shift depends on, the four parameters: average elements height ($K_{avg}$), the area projected on the surface ($A_p$), area projected in flow direction ($A_s$), and the average distance between elements ($L_{avg}$) (as shown in Fig. 1) are contracted together to one dimensionless parameter called equivalent sand roughness height ($K_s$) that can be calculated by the equations presented in Dirling (1973)

$$\frac{K_s}{K_{avg}} = \begin{cases} 0.0164 \times \Lambda^{3.78} & \text{for} \quad \Lambda < 4.93 \\ 139.0 \times \Lambda^{-1.90} & \text{for} \quad \Lambda > 4.93 \end{cases} \tag{4}$$

where

$$\Lambda = \frac{L_{avg}}{K_{avg}}\left(\frac{A_p}{A_s}\right)^{-4/3} \tag{5}$$

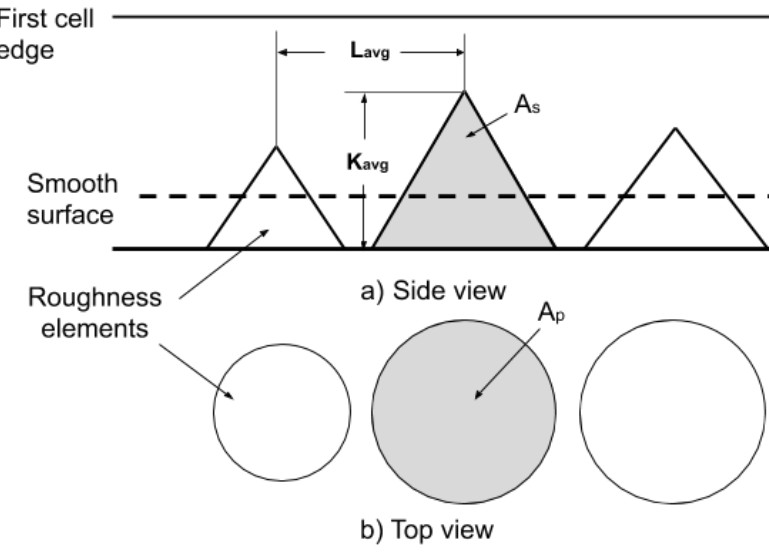

**Figure 1.** Roughness element geometrical parameters

So, once the roughness profile is known, $K_{avg}$, $L_{avg}$, and $D_{avg}$ can be calculated. Assuming roughness elements take certain geometrical shape for approximation, $A_p$ and $A_s$ can be calculated.In the current study, roughness elemnts are assumed to take





conical shapes. Accordingly, $A_p/A_s = \pi D_{avg}/2K_{avg}$ and Equations 4 and 5 can be implemented to calculate $K_s$. The final value of the velocity shift in all RWFs used in this work depends on the calculated value of $K_s$.

### 2.2.1 Momentum RWF

Da Silva et al. (2011) applied the velocity shift to the well-known log-low equation by introducing a new term, namely $\Delta B$, that represents this velocity shift as a function of $K_s$ according to Cebeci (2004) ideas in the equations

$$u^+ = \frac{1}{\kappa}\ln(Ey^+ - \Delta B) \tag{6}$$

$$\Delta B = \begin{cases} 0 & : K_s^+ \leq 2.5 \\ \frac{1}{\kappa}ln[K_s^+ - \frac{2.25}{87.75} + C_s K_s^+]\sin[0.4258(\ln K_s^+ - 0.811)] & : 2.5 < K_s^+ < 90 \\ \frac{1}{\kappa}ln(1 + C_s K_s^+) & : K_s^+ \geq 90 \end{cases} \tag{7}$$

Where $y^+ = y(1)u_\tau/\nu$, $K_s^+ = u_\tau K_s/\nu$ , $E = 9.8$ , $\kappa = 0.41$ and

$$C_s = \frac{E}{32.6} - \frac{1}{K_s^+} \tag{8}$$

Finally, turbulent viscosity can be calculated by:

$$\nu_t|_w = \nu\left(\frac{y^+\kappa}{\ln(Ey^+/e^{\kappa\Delta B})}\right). \tag{9}$$

Since this wall RWF deals mainly with $\nu_t$, it is used with Spalart-Allmaras turbulence model to simulate rough surface.

### 2.2.2 DLR RWF

In their work, Knopp et al. (2009) followed the ideas of Aupoix and Spalart (2003) in adding the velocity shift effect to the wall function. However, instead of applying the shift on turbulent viscosity as in Aupoix and Spalart (2003), they used their procedure to adapt k and $\omega$ turbulence parameters. The new k and $\omega$ RWF states that

$$\omega|_w = min\left(\frac{u_\tau}{\beta_k^{1/2}\kappa\tilde{d}_o}, \frac{60\nu}{\beta_\omega y(1)^2}\right) \tag{10}$$

$$\tilde{d}_o = 0.033\phi_{r_2}k_s \tag{11}$$

$$\phi_{r_2} = min(1, \frac{k_s^{+2/3}}{30})min(1, \frac{k_s^{+1/4}}{45})min(1, \frac{k_s^{+1/4}}{60}) \tag{12}$$

$$k|_w = \phi_{r_2}k_{rough} \tag{13}$$

$$k_{rough} = \frac{u_\tau}{\beta_k^{0.5}}, \phi_{r_2} = min(1, \frac{k_s^+}{90}) \tag{14}$$

Where $\beta_k = 0.09, \kappa = 0.41, \beta_w = 0.075$. Unlike Momentum RWF explained in section 2.2.1, this RWF deals with modifying k and $\omega$ values near the wall. Hence, this RWF is used with k-$\omega$ SST turbulence model.



### 2.2.3 Colebrook RWF

Chedevergne and Aupoix (2017) applied Suga et al. (2006) approach in adapting k and $\omega$ turbulence parameters to rough

surfaces and fitted the resulting formula to Colebrook's experimental data. This resulted in:

$$k|_w = max\left(0, \frac{1}{\sqrt{\beta^*}}\tanh\left[\left(\frac{\ln\frac{k_s^+}{90}}{\ln 10} + 1 - tanh\frac{k_s^+}{125}\right)tanh\frac{k_s^+}{125}\right]\right) \tag{15}$$

$$\omega|_w = \left(\frac{300}{k_s^+}\left(tanh\frac{15}{4k_s^+}\right)^{-1} + \frac{191}{k_s^+}\left[1 - exp\left(-\frac{k_s^+}{250}\right)\right]\right) \tag{16}$$

Also the same like DLR RWF, this one is used with k-$\omega$ SST turbulence model.

## 3 Profiles and grids of CFD simulations

To enable the usage of the aforementioned RWF's, rough ice profiles have to be smoothened to give an equivalant smooth
surface. By using this smooth surface and the $K_s$ value calculated from equ. 4, the RWF's can be used in the simulation cases.
This section explains how these profiles should be prepared and how the computational grid should be generated.

### 3.1 Wind turbine airfoil with ice accretion

To investigate the different wall functions described in Section 2, they were applied on a wind turbine airfoil with two different
ice accretion profiles. The two ice profiles were collected from site, scanned and then machined as a leading edge attachment
for wind tunnel measurements.

Both ice profiles take the horn-ice form Fig. 2 which usually occur under severe icing conditions and also both of them extend
to 7.5-10 % of chord length. However, it can be noticed that profile 1 is smoother than profile 2. Also it can be noticed that
profile 1 takes a more aerodynamic shape than profile 2 since profile 2 has two horns which forms a stagnation area between
them.

### 3.2 Roughness parameters calculations

Since all RWFs treat the rough wall as smooth wall with a velocity shift as explained in 2.2, the actual rough surface of the ice
profile is replaced with another equivalent smooth surface. The new smooth surface will be used to generate the computational
grid around the profile and will be numerically treated as rough surface, i.e. a velocity shift will be added to the smoothened
surface. To calculate this new surface, the rough surface was smoothened with cubic splines as shown in Figures 3a and
3b. Knowing the distance between roughness elements and height of elements, average sand roughness height Fig.1 can be
calculated using Eq. (4) assuming roughness elements take conical shapes, then $A_p = \pi D_{avg}^2/4$ and $A_s = 0.5K_{avg}D_{avg}$.
The above analysis gives results in $K_s = 1 \times 10^{-3}m$ and $1 \times 10^{-2}m$ for profiles 1 and 2 respectively.



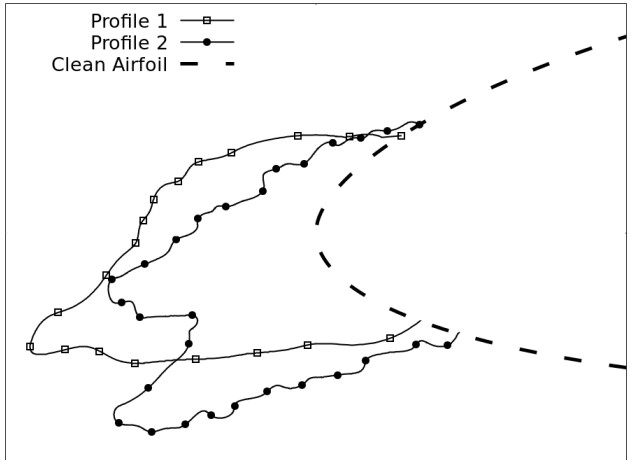

**Figure 2.** Ice profiles 1 and 2

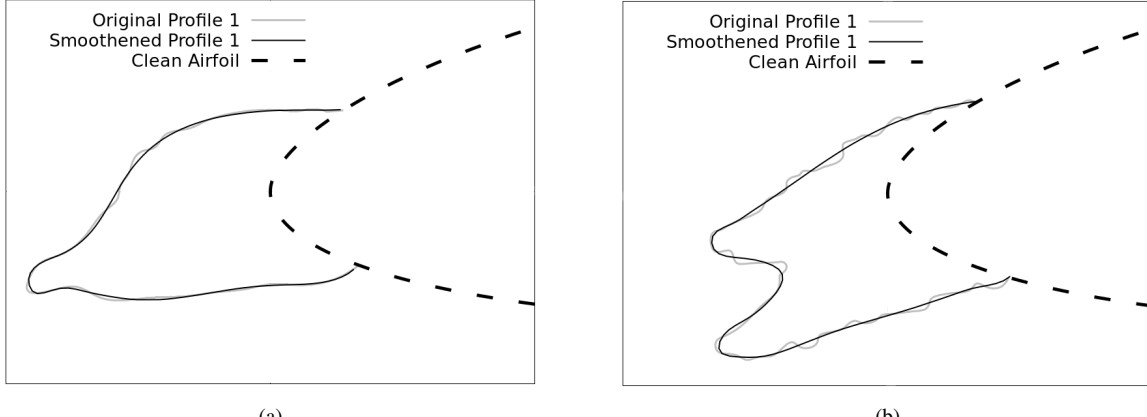

**Figure 3.** Comparison between original and smoothened ice surfaces for: (a) profile 1 and (b) profile 2

### 3.3 Grid generation

In order to use rough wall functions, the height of the first cell center should be large enough to cover the roughness element. Along with converting the rough surface into a smoother one, the resulting grid is much coarser than the grid required by smooth wall functions which require $y^+(1) < 1$ to be able to correctly simulate the boundary layer. Accordingly, the studied approach in this work requires less computational resources. In case of the two ice profiles studied in this work, to properly generate a grid that fulfills the condition of $y^+(1) < 1$, is found to require number of cells $\approx 4 \times 10^5$ cells in 2D simulations.

On the other hand, when using rough wall functions with the proper first cell height and roughness smoothing, it is found to



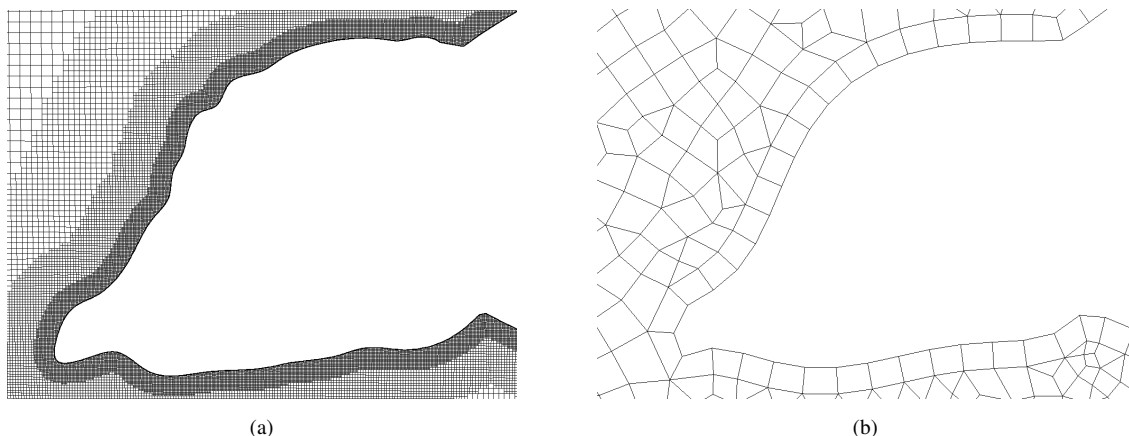

| (a) | (b) |
|---|---|

**Figure 4.** Comparison between: (a) fine and (b) coarse grids of profile 1

require $\approx 1 \times 10^5$ cells in 2D simulations. A comparison between smooth and rough wall function grids is shown in Figures 4 and 5. Resulting from the numerical setup described above, the computational grids used in this work are given in Tab.1.

**Table 1.** Simulation cases

| Case No. | Profile | Re | y(1) [m] | AoA [$^o$] |
|---|---|---|---|---|
| 1 | profile 1 | $2.6 \times 10^6$ | $9 \times 10^{-3}$ | -4$^o$ to 16$^o$ |
| 2 | profile 1 | $5.2 \times 10^6$ | $9 \times 10^{-3}$ | 0$^o$ to 16$^o$ |
| 3 | profile 2 | $3.1 \times 10^6$ | $10 \times 10^{-3}$ | -4$^o$ to 16$^o$ |

## 4 Simulation results

For simulation results, validation case will be introduced to give evidence of correct implementation of the new codes. Then,
the results of the simulation cases shown in Tab.1 will be shown.

### 4.1 Implementation and validation

To apply the aforementioned RWF, equations 10-16 were implemented within OpenFOAM framework. The Momentum RWF is already available in OpenFOAM (named nutURoughWallFunction). However, the other two wall functions, namely DLR and Colebrook RWF, are not available. These two wall functions were implemented and validated against experimental re-
sults published by Achenbach (1977). In this paper, the author analyzed the flow field around a circular copper cylinder with paramedical roughness elements to study both fluid flow and heat transfer. This experiment was done on a 0.5 m length, 0.15




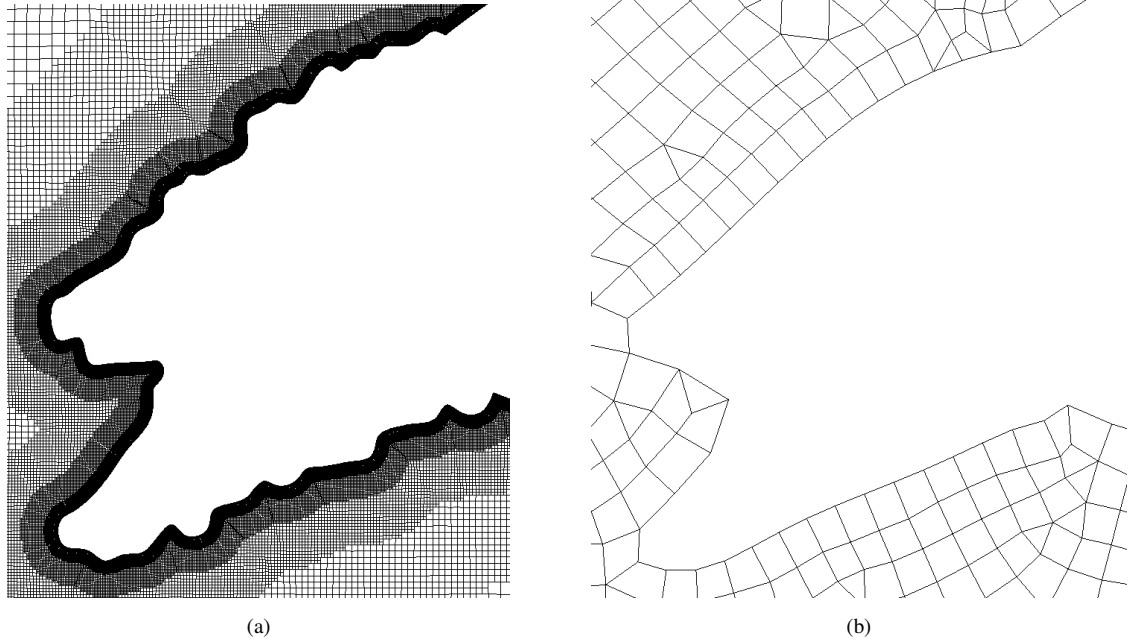

(a)          (b)

**Figure 5.** Comparison between: (a) fine and (b) coarse grids of profile 2

m diameter copper cylinder at different roughness heights and Reynolds numbers.

It is known that RANS simulations of flow filed around a cylinder is already unstable and hard to predict, especially when it comes to the prediction of separation location. However, this case was selected to identify any false implementation of RWFs in OpenFOAM that may lead to unstable numerical solution. Figures 6-a, b and c shows the results of simulating circular cylinder at $Re = 4 \times 10^6$ and $K_s = 75 \times 10^{-5}$ m, $3 \times 10^{-3}$ m, $9 \times 10^{-3}$ m, and smooth cylinder surface respectively. Despite the experiments were carried out at different $Re$ numbers, this work only considered the comparison with $Re = 4 \times 10^6$ since this value is in the same order of magnitude of the wind tunnel results of iced airfoils that will be shown in the next sections. Also, this work didn't take into consideration the heat transfer results of Achenbach. The main focus was the fluid flow only.

As shown in figures 6-a, b and c, all on the three RWFs managed to predict $C_p$ of this case correctly between $0^o$ and $60^o$ of the cylinder surface. For angles higher than $60^o$, each RWF had different behavior. For the Momentum RWF, a good agreement between numerical and experimental results can be predicted up to $100^o$ where the model fails to correctly simulate the separation location. On the other hand, both DLR and Colebrook RWFs underestimate the value of $C_p$ and overestimate the separation location over the cylinder.

Figures 6-a, b and c also show the results of simulation of smooth, resolved cylinder simulation using Splart-Allmaras Turbulence Model. In comparison with Momentum RWF simulations, it can be noticed that the RWF caused earlier prediction of the location of separation. This matches with the fact that rough surface prevents transition from laminar to turbulent flow and initiates separation earlier.

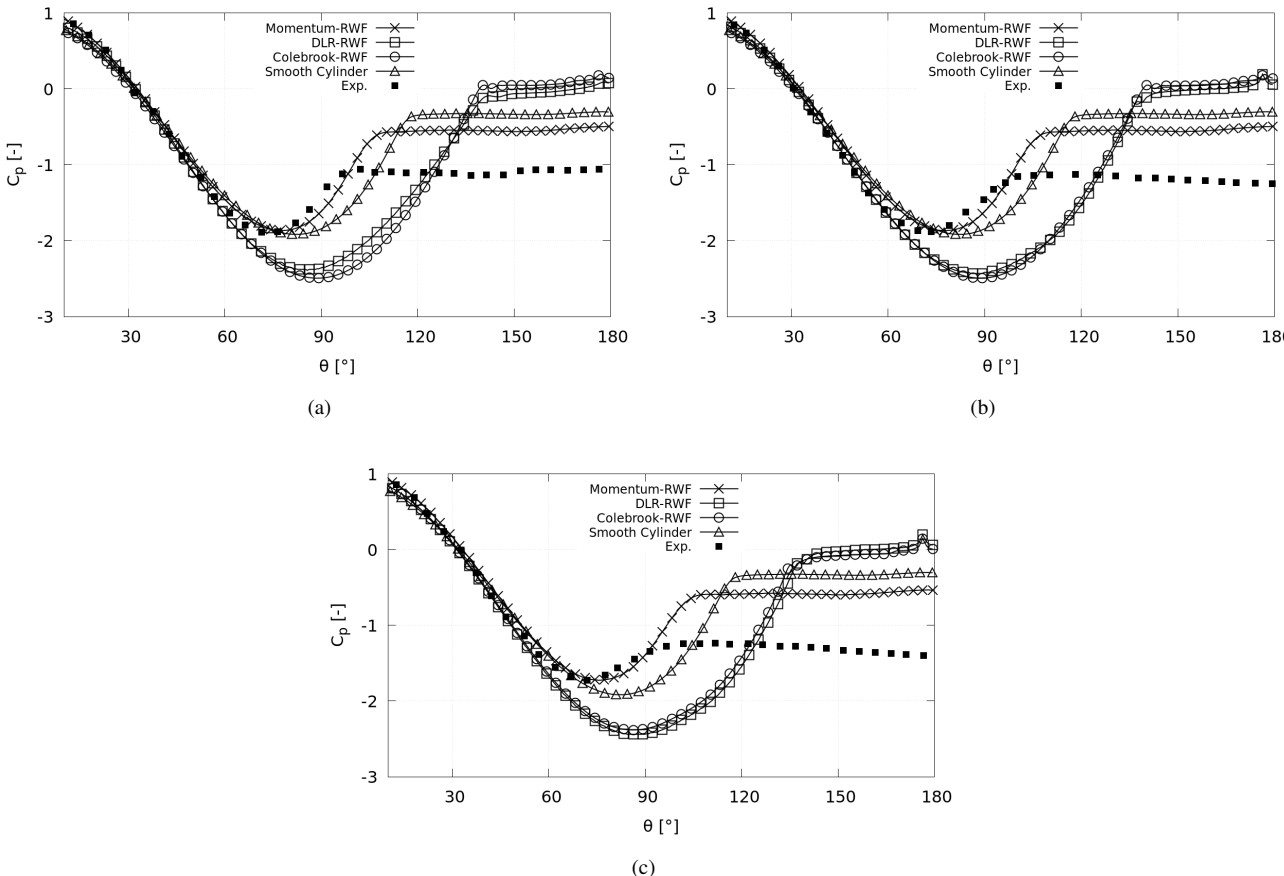

**Figure 6.** $C_p$ distribution over rough cylinders using implemented wall functions at $Re = 4 \times 10^6$ and: (a) $K_s = 75 \times 10^{-5}$ m, (b) $K_s = 3 \times 10^{-3}$ m, (c) $K_s = 9 \times 10^{-3}$ m where $\theta = 0^o$ at stagnation point and increases downstream.

While analyzing the results of these validation cases, one must keep in mind that such a complicated simulation case of flow
over a cylinder at high Reynolds number using steady-state RANS simulation is difficult. Accordingly, one should not expect
good agreement between numerical and experimental simulation on all locations over the cylinder. These low expectations
comes from the fact that all RANS models fail to accurately predict the flow separation location even for airfoils with relatively
high AoA. For cylinder case, the situation is even harder. Since the aim of this validation process is to make sure that the RWFs
have been correctly implemented and are working properly and not to judge the mathematical behavior of these RWFs, it can
be concluded that the RWFs follows the trend of $C_p$ distribution over the cylinder.

### 4.2 Case 1: profile 1 at Re = 2.6 × 10⁶ :

In this case, profile 1 was tested at Re = $2.6 \times 10^6$, which is a relatively low Reynolds number compared to the other two cases.
Fig.7 shows the comparison between the predictions of each of the three RWFs with wind tunnel results. On the other hand,



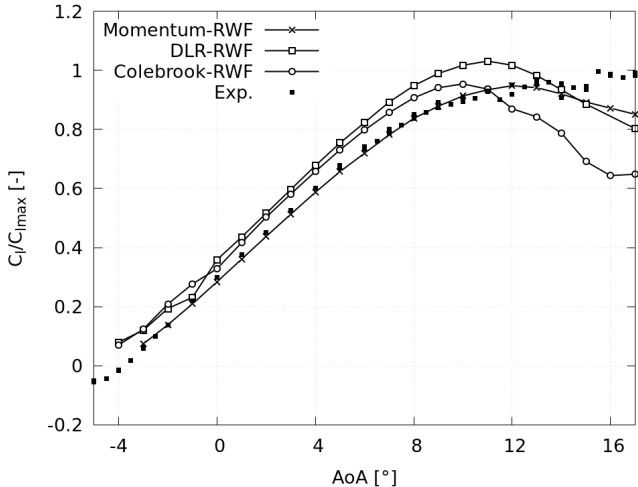

**Figure 7.** Case 1: $C_l$ vs AoA for different wall functions

figures 8a-8d gives an deeper look at the prediction of pressure distribution on the surface of the profile and compare the results
with pressure-tap measurement values.

It can be noticed from fig.7, and also expected, that each model has a different prediction for the maximum lift coefficient
$C_{l_{max}}$ value or the angle of attack (AoA) at which this $C_{l_{max}}$ occurs. However, Momentum-RWF showed the best agreement
with $C_l$ values. On the other hand, the pressure coefficient ($C_p$) shows different behaviors depending on the angle of attack.
Figures 8a-8d show that the $C_p$ distribution at AoA = $0^o$, $4^o$, $8^o$, and $12^o$ respectively. At AoA = $0^o$, a good agreement be-
tween experimental and CFD results can be seen except for the lower surface region at x/c $\approx$ -0.05 to 0.05. At AoA's = $4^o$
and $8^o$, good agreement was achieved over both smooth and rough surfaces of the airfoil. This agreement starts to suffer from
some deviations at AoA = $12^o$. These deviations are expected due to relatively high AoA that is higher than the AoA of max $C_l$.

### 4.3 Case 2: profile 1 at Re = $5.1 \times 10^6$ :

This case is exactly like the previous case except for being tested at Re = $5.1 \times 10^6$. This high Reynolds number is challenging
for RWFs since it leads to more violent separation and hence are harder to be predicted.
This case shows good agreement (Fig.9) especially in the linear region of AoA vs $C_l$ relationship. However, over prediction of
$C_{l_{max}}$ can be noticed in DLR and Colebrook RWFs results.
The effect of the high Reynolds number can be seen in $C_p$ distribution curves (Figures10a-10d) where large differences between
DLR RWF predictions and experimental results occurs for AoA = $0^o$ and $4^o$. Also Colebrook RWF shows under estimation
of pressure on the upper surface of the airfoil at AoA = $4^o$. While the Momentum rwf shows better agreement for all studied





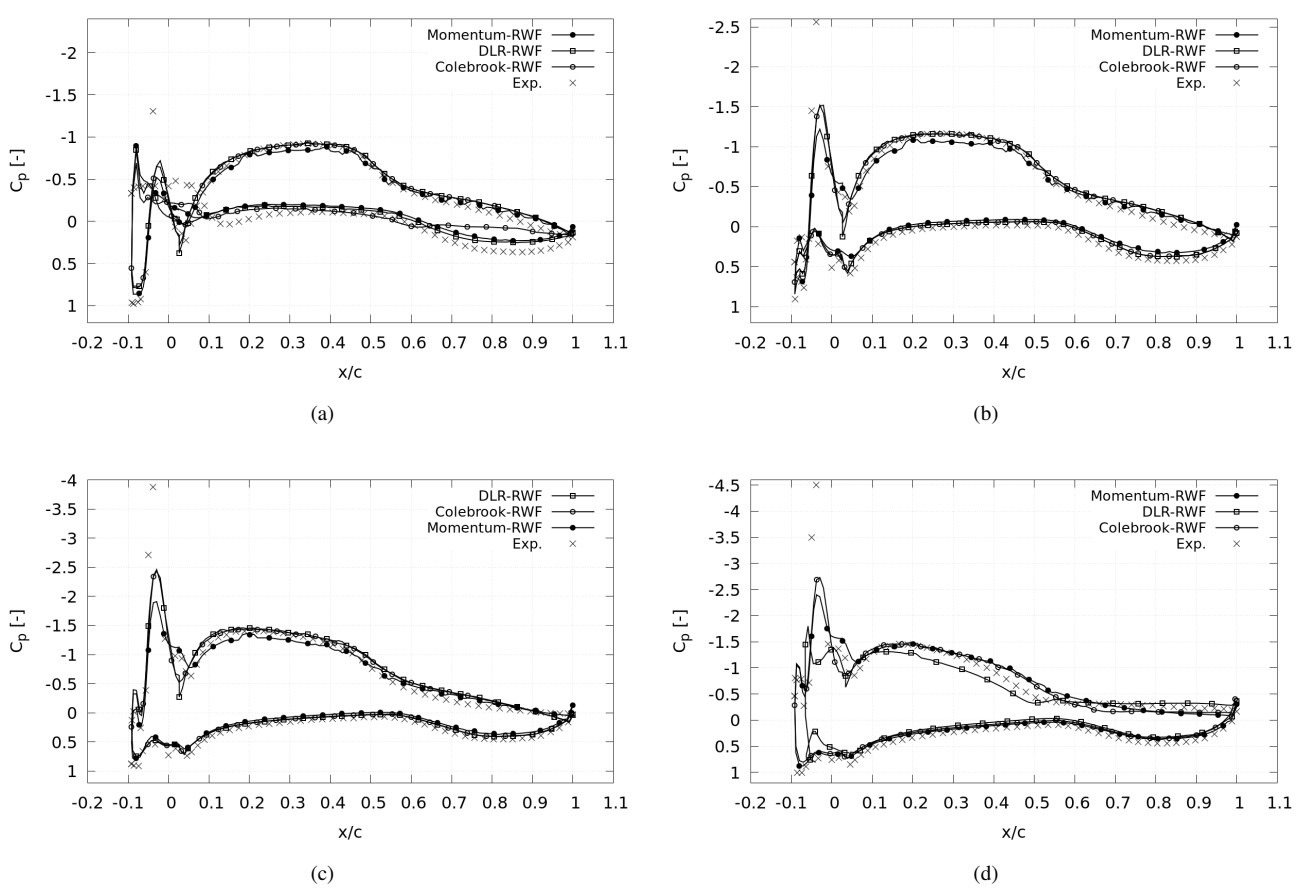

**Figure 8.** Case 1: $C_p$ distribution for AoA = (a) $0^o$, (b) $4^o$, (c) $8^o$, and (d) $12^o$

AoA's.

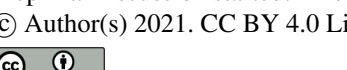



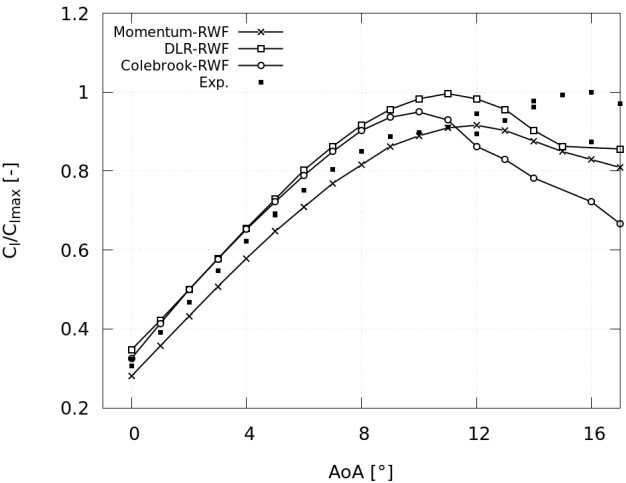

**Figure 9.** Case 2: $C_l$ vs AoA for different wall functions

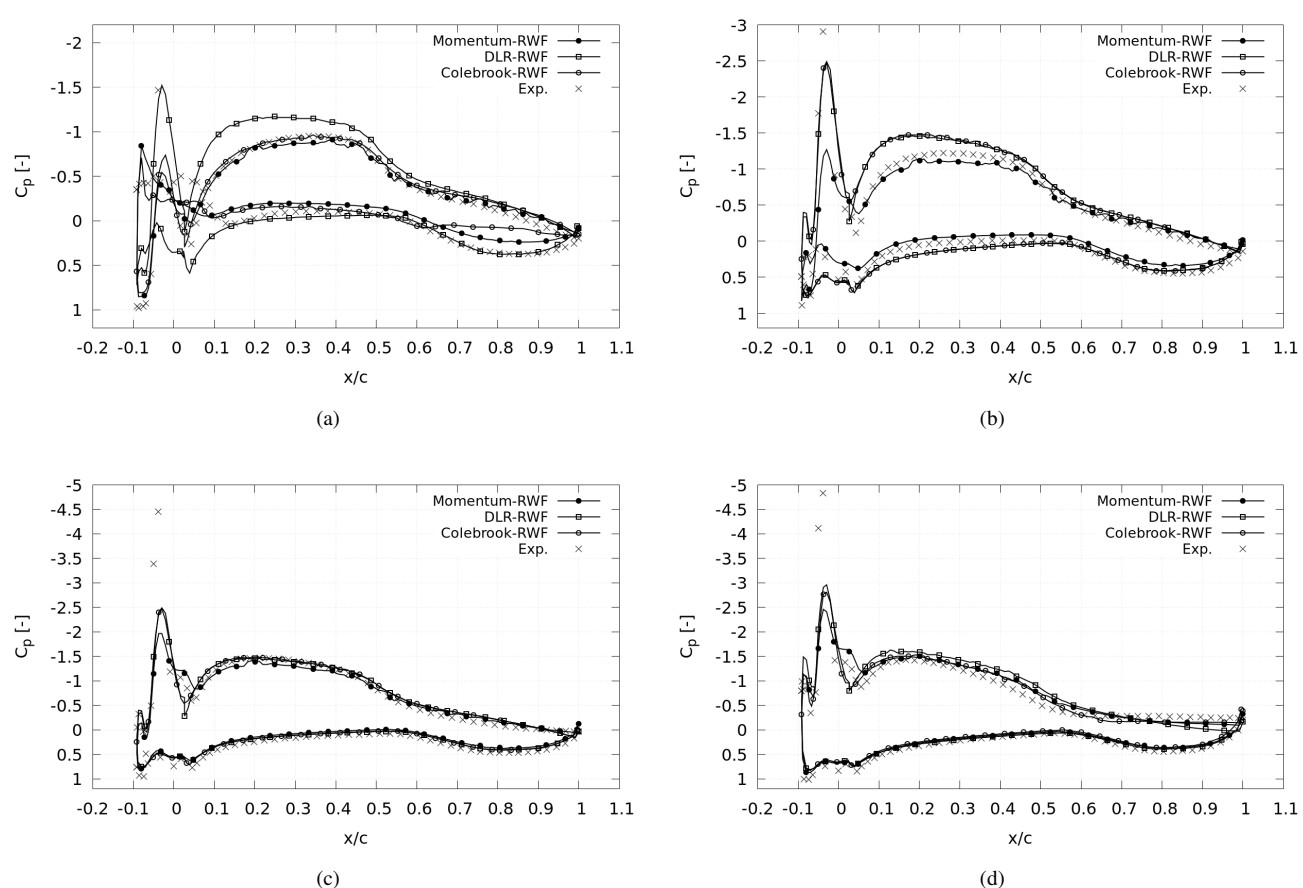

**Figure 10.** Case 2: $C_p$ distribution for AoA = (a) $0^o$, (b) $4^o$, (c) $8^o$, and (d) $12^o$



## 4.4 Case 3: profile 2 at Re = $3.1 \times 10^6$ :

In this case, the flow over profile 2 was simulated. It can be noticed from Fig.2 that profile 2 has not only two ice horns, but also it has a rougher surface , i.e. has a higher $K_s$ value. The effect of this complex shape can be obviously seen Fig. 11. It can be noticed from this figure that the $C_l$ curve does not show a clear stall AoA. Only the slope of the $C_l$ curve decreses starting from AoA = $5^o$.

Fig.11 also shows that all compared models had good agreement with experimental results for AoA's in range between $-1^o$ and $5^o$. Out of this range, each model shows different behavior. For Colebrook and Momentum RWFs shows large deviations from experimental results for AoA's higher than $5^o$ while DLR RWF shows better agreement in this range. For $C_p$ distribution, Fig.12a shows good agreement between experiment and all simulations at AoA = $0^o$ on the upper surface of the airfoil while they have higher deviations from experiment on the lower surface in the x/c range between -0.1 to 0.4. At AoA = $4^o$ results shown in Fig.12b, Colebrook and DLR RWFs shows large deviations over the whole airfoil while Momentum RWF shows better agreement. In Figures 12c and 12d good agreement between all models and experimental results can be noticed.

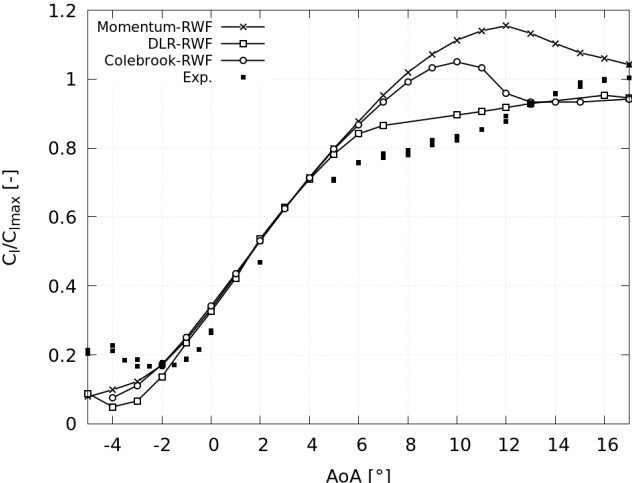

**Figure 11.** Case 3: $C_l$ vs AoA for different wall functions





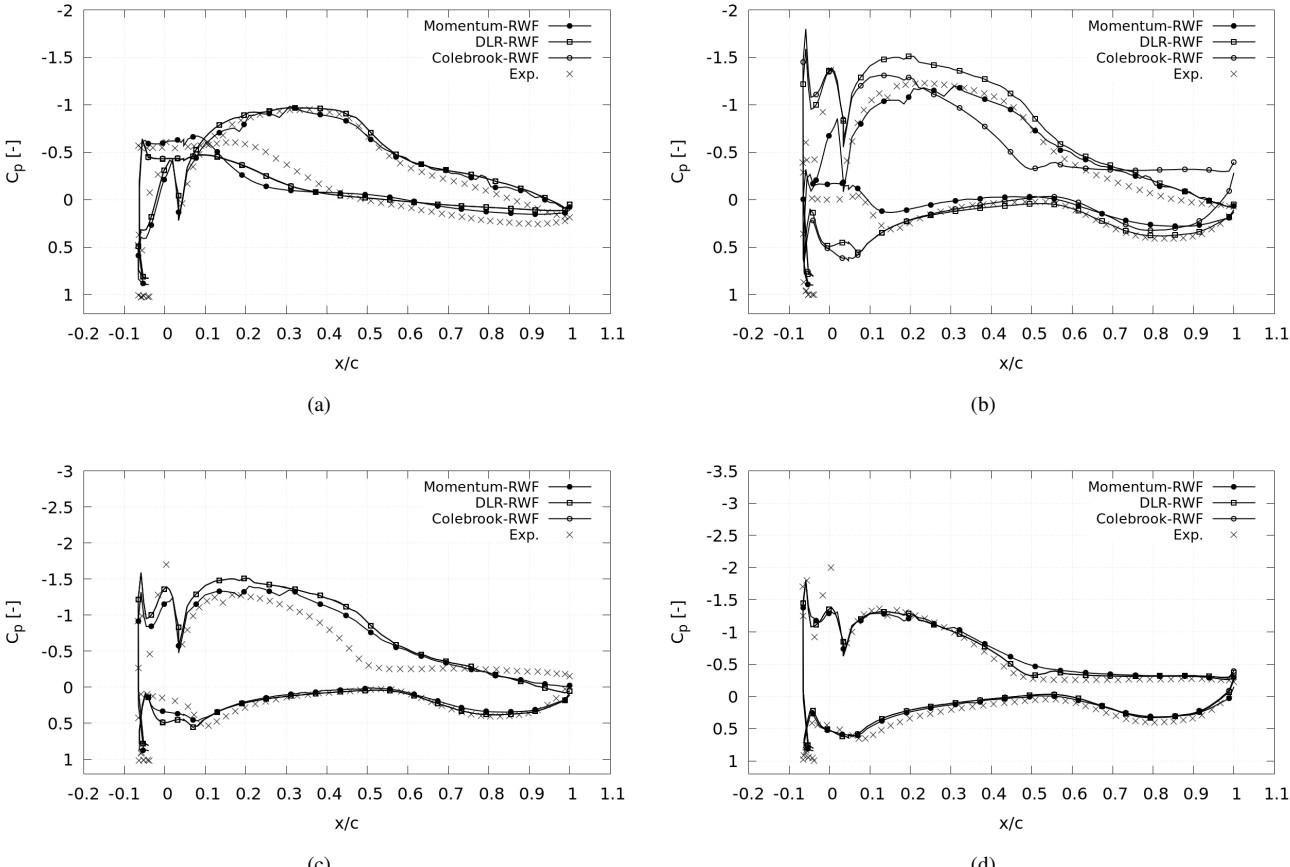

**Figure 12.** Case 3: $C_p$ distribution for AoA = (a) $0^o$, (b) $4^o$, (c) $8^o$, and (d) $12^o$

## 4.5 Agreement analysis

The results shown in the last sections show that the flow field has different behaviors with using each RWFs. To find out which of the three RWFs resulted in the most accurate results, error analysis should be done to find out which RWF had less deviation

from the experimental results. $C_p$ distribution was chosen as the main criteria to compare since as explained earlier, the $C_p$ distribution gives a better understanding of the shape of the flow field over the body which is very important to simulate ice accumulation. In this work, the average absolute error between $C_p$ calculated from pressure measurements in wind tunnel and the corresponding $C_p$ calculated from simulations using the equation:

$$e_{avg} = \frac{1}{N} \sum_{1}^{N} |C_{p,exp} - C_{p,sim}| \tag{17}$$

where $e_{avg}$ is the average error and $C_{p,exp}$ and $C_{p,sim}$ are coefficients of pressure of experiments and simulations respectively. As shown in the figures 13a and 13b, it can be noticed that the higher the AoA value, the more error between simulation and



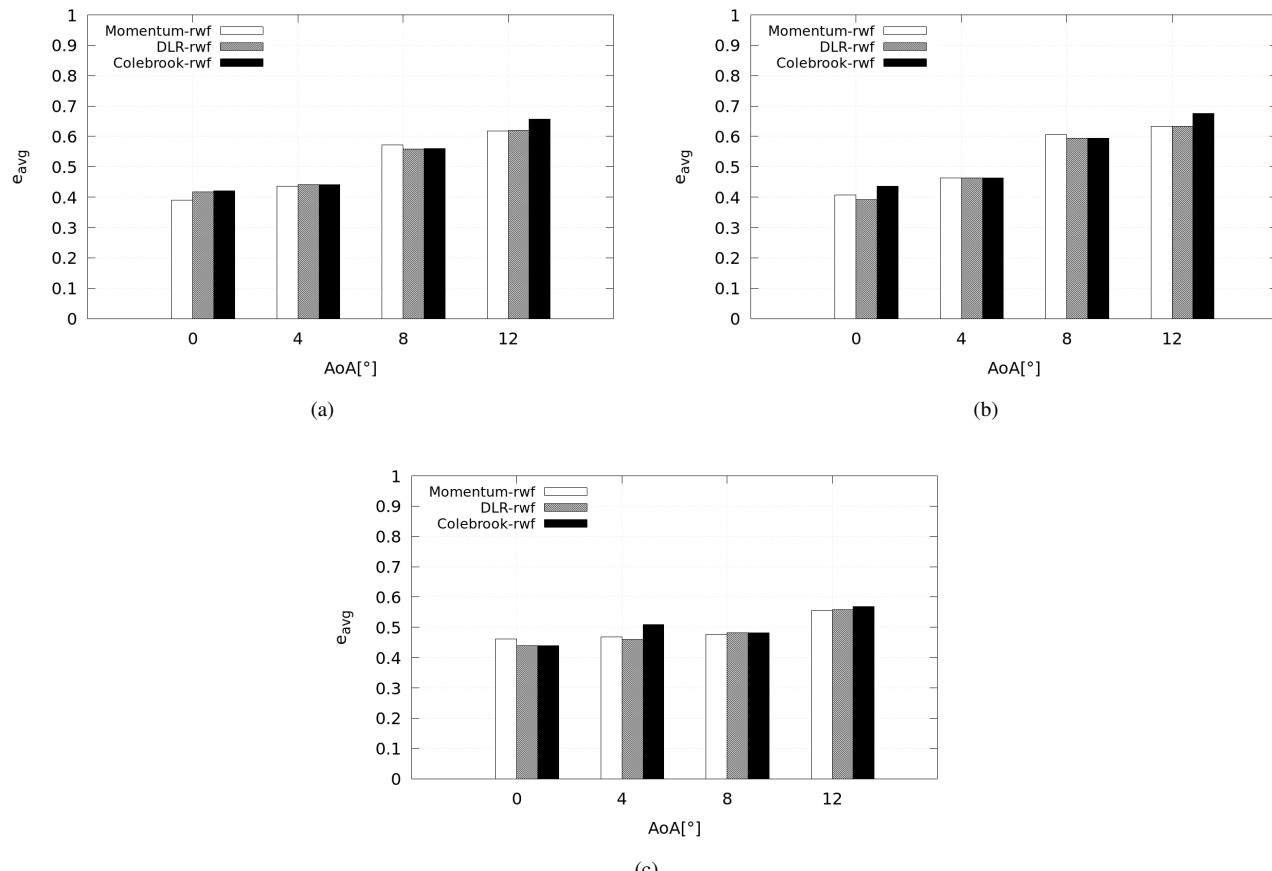

**Figure 13.** Average error between experimental and simulation results of $C_p$ distribution (a) case 1, (b) case 2, and (c) case 3

wind tunnel results. While in Fig. 13a, we can see that the error values are not increasing with the increase of AoA in the same rates as shown in the previous two figures. These error results shows that all RWFs has a limited capabilities in simulating the detachment of the flow and accordingly give results deviated from actual results. Also, in case 3, where profile 2 was simulated,

the detachment and reattachment of the flow was more violent at all tested AoA's that it caused close error values between all cases except for AoA = $12^o$ that showed higher errors.

## 5 Discussion and conclusions:

In this work, three different rough wall functions were tested on iced wind turbine airfoils. The aim of this comparison was to find out which RWF will be the most suitable one to simulate ice accretion on wind turbine blades exposed to icing atmospheric

conditions. To be able to apply these RWFs, DLR and Colebrook RWFs were implemented to OpenFOAM v6 CFD framework along with the existing Momentum RWF. After that, two ice profiles that collected from wind site, molded to airfoil, and





tested in the wind tunnel were smoothed by a cubic spline to find the equivalent smooth surface. Also, roughness parameters described in Section 3.2 were calculated and used to calculate the velocity shift value $\Delta u$.

In this section, general remarks and discussions of the results are introduced. Then the conclusions from the outcomes of this
work will be highlighted.

### 5.1 General remarks and discussion

Regarding the rough ice surface shown in Fig. 2, one should keep in mind that the ice formation process (or roughness formation in general) is a stochastic phenomenon. Which means, if the same airfoil exposed to the same atmospheric conditions, the ice profile resulting will not be exactly the same. So, to find the real smooth surface required for the simulations, a large number
of ice profiles of the same atmospheric conditions should be studied and averaged to find the real average surface. Also, each rough surface will give different attachment and reattachment bubbles' locations, other than what is shown in Fig. 14, and might have different overall pressure distribution. However, the scope of this work is only to prove that the RWF approach results in good results compared to experiments.

From the geometry of the two ice profiles shown in Fig.2, it can be noticed that both profiles are slightly inclined downwards.
This inclination forms relatively big separation bubble behind the profile on the lower surface as shown in Fig. 14. Thats is why we can see deviations between simulations and experimental results of $C_p$ distribution in this region (i.e. the lower surface between x/c = -0.1 to 0.1) in most of the studied cases.

It can be noticed from the results that agreement of $C_l$ curves between simulations and experimental results doesn't necessarily mean that flow field was accurately simulated. For example, in case 3, DLR RWF shows better $C_l$ agreement than other models
at AoA = $4^o$ while from $C_p$ distribution, DLR RWF showed larger deviations from experimental results. The same happened with case 2 at AoA = $0^o$ and $4^o$. The overall $C_l$ at these AoA's in the end had good agreement because the deviations on the upper and lower surfaces compensate each other which can be misleading in this case. To accurately access the $C_l$, it should be studied together with the $C_p$ distribution curves.

It is noteworthy that all the RWFs used are only algebraic equations that express the behavior of the flow near that wall.
Accordingly, the three different implementations have the same computational cost and the overall computational cost depends only on number of cells in the used computational grid.

### 5.2 Conclusions

The simulation cases shows that Momentum RWF provides the best agreement of simulations with experiments compared to the other models. However, in some cases, the simulations show fair agreement at locations that witness violent separation like
concave areas between ice and airfoil. This fair agreement is expected due to using steady state RANS simulations. Also, it can be concluded that the DLR RWF also gives good agreement in most of the cases and was close to results achieved by the Momentum RWF.

From this work, we can conclude that in simulations of rough, iced airfoil profiles, the Momentum RWF should be used to simulate the effect of the presence of roughness on the overall aerodynamic performance. Such method should be beneficial



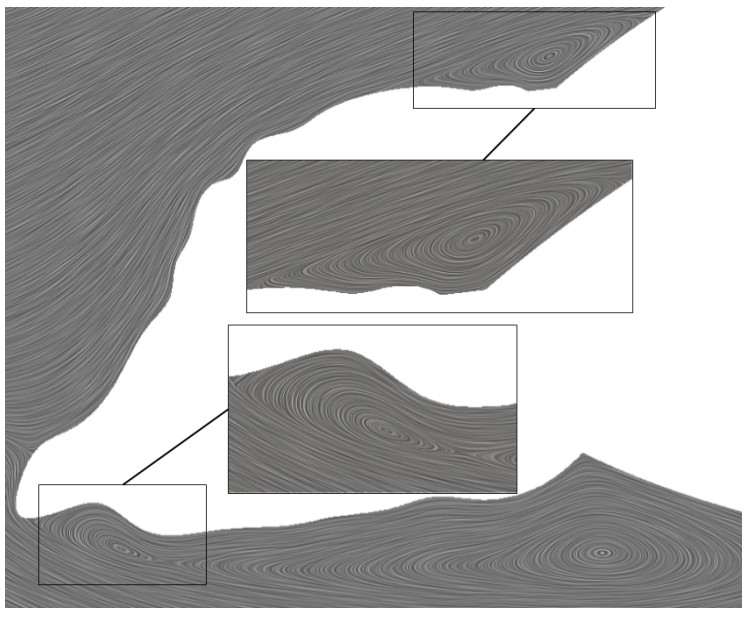

(a)

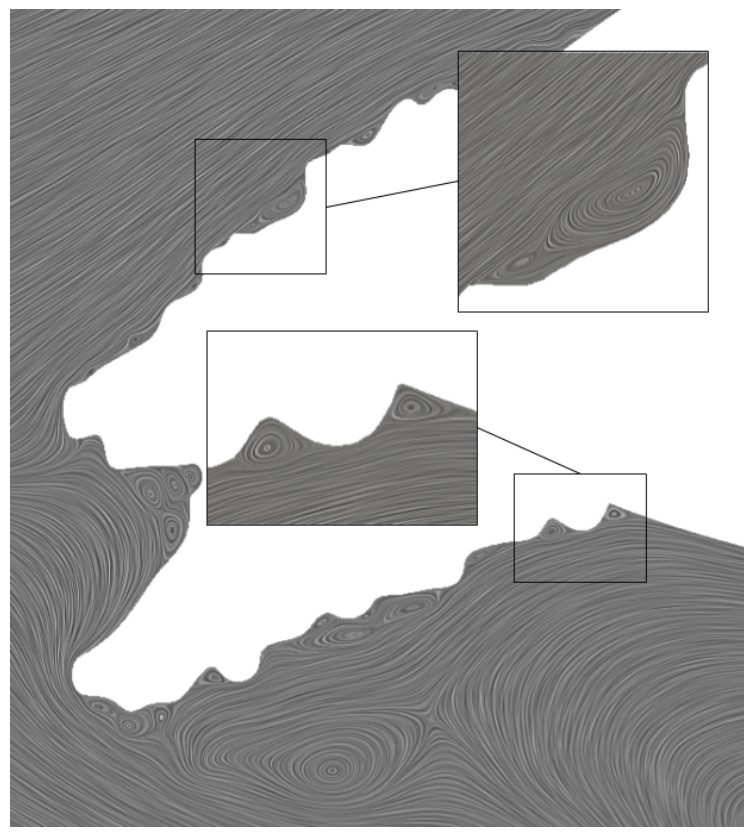

(b)

**Figure 14.** LIC of flow in (a) case 2 at AoA = $0^o$ (b) case 3 at AoA = $0^o$ with zoom-in on some local separation locations





for the simulation of the performance drop of wind turbine blades exposed to icing atmospheric conditions with minimum computational effort.

## Appendix A: Nomenclature

| | | |
|---|---|---|
| $A_p$ | Area projected normal to streamwise direction | **Abbreviations** |
| $A_s$ | Area projected to streamwise direction | RWF  Rough wall function |
| $\Delta B$ | Log-law velocity shift | RANS  Reynolds-averaged Navier-Stokes |
| $C_l$ | Coefficient of Lift | **Greek letters** |
| $C_{l_{max}}$ | Max lift coefficient | $\kappa$  von Karman constant |
| $C_p$ | Coefficient of pressure | $\omega$  Specific rate of dissipation |
| $D$ | Roughness element diameter | $\nu$  Eddy viscosity |
| $k$ | Turbulent kinetic energy | **Subscripts and superscripts** |
| $K$ | Element Roughness Height | $avg$  average value |
| $K_s$ | Equivalent Sand Roughness Height | $exp$  experimental result |
| $L$ | Distance between two roughness elements | $sim$  simulation result |
| $u_\tau$ | Friction velocity | $w$  wall value |
| $y(1)$ | First cell height | $+$  value in wall scaling |

*Author contributions.* K.Y. simulated the cases and wrote the article. H.K. and K.Y. implemented methods in OpenFOAM. T.K. provided
experimental data and consultation regarding experimental results. B.S. coordinated project. B.S. and J.P. provided consultation and supervision.

*Competing interests.* The authors declare no conflict of interest.

*Acknowledgements.* This work is part of project OptAnIce: Optimales Anti-Icing für Rotorblätter im kalten Klima, funded by the German
Federal Ministry for Economic Affairs and Energy (BMWi), fund No.: 0324232C. The simulations were performed using the HPC Cluster
EDDY in the University of Oldenburg, a part of WIMS-Cluster project and funded by the BMWi.



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
