# Peer review of "Numerical Investigation of Aerodynamic Performance of Wind Turbine Airfoils with Ice Accretion"

_Wind Energy Science, 2021_

## Referee Comment (RC1)

Sections 2.1.1 and 2.1.2. I don't think it's necessary for the authors to explain the turbulence models used: these are well known to readers experienced in CFD and can be researched by the interested reader in provided references if necessary. It would be preferable to briefly go over advantages and disadvantages of each model and to elaborate on why they were used in this specific case. Also the choice of using two turbulence models in the same study seems odd, so further explanation on this aspect should be provided.

Figure 2: The scanned data points shown in figure 2 for the two ice-profiles appear to be quite far apart. From what is stated in section 3.2 the scanned profile is used to calculate the average sand grain roughness. Please justify the expected impact of the number of scanned points on the modelling.

Section 3.2: How can one differentiate between roughness and geometry? In other words what part of the geometry can be justifiably smoothed and included in the rougness wall functions and what part must be retained? A robust criterion is not provided here. See final comments for further remarks on this.

Section 3.2: Assuming roughness elements with a conical shape seems quite semplicistic. Please justify this choice

Section 3: Are rough wall functions used on the entire airfoil or just on the iced part? If it's the first, how does using rough wall functions on the entire airfoil influence the results? Perhaps it would be useful to check that all the wall functions proposed give reliable and consistent results for an airfoil with no icing.

Figures 4 and 5: Please explain these figures better. If the "fine" and "coarse" grid refer to cases where wall functions are used vs cases where they are not the authors must explain the difference in the size of the elements surrounding the airfoil. The coarse grid differs not only for the boundary layer but also in the flowfield. In my opinion this can significantly influence the results.

Figures 4 and 5: The meshes appear different in the fine and coars cases. "Fine" seems to use Cartesian cut-cell meshing, coarse uses polygonal cells. Please explain the differences.

Section 3.3: Was a mesh independence study performed?

Table1 : Why different Reynolds numbers? I suggest including the total number of elements of the mesh aswell.

Figures 7,9 & 11: The "Exp." Dataset appears to have duplicate data points for some AoAs. Fpr instance Fig. 9 AoA=12, 2 "Exp." Points are clearly visible. Please explain. Also, why are this figures relative to maximum lift? This could accentuate or diminish the difference between models depending on the value of maximum lift.

Section 4.4: As stated previously it is hard to judge differences in lift given the relative nature of figure 11. However, as it stands the differences seem pretty substantial even in the linear region.

Figure 12: here seems to be a considerable difference in figure 12b between momentum model and the others. This is not reflected in figure 11 at AoA=4°, why?

Line 240: This phrase is not clear, please revise it.

Section 5.1: Authors state that many icing profiles should be measured and averaged to get a good reference shape of the airfoils. What about the roughness values? How can those be estimated and used in computations?

Line 260: I suggest to edit figure 12 to include all of the tested roughness models. This would allow to compare if and where separation is predicted. Also, separation in this case seems like it is a consequence of the shape of the ice and not relative to roughness. If a model cannot correctly predict this separation can this be caused by excessive geometry smoothing? In other words, could agreement be improved is a different smoothing strategy was adopted? Or if no smoothing was adopted and roughness height measured differently?

Line 263: This is a good point. What kind of wall function is used in the non-iced part of the airfoils?

---

## Referee Comment (RC2)

[referee-annotated manuscript omitted]

---

## Author Comment (AC1)

**Responses to Reviewer Comments: Report #1**

*We would like to thank the reviewer for the effort in reviewing and commenting on this article. The comments provided have helped to improve the manuscript considerably.*

*Please find below our responses to each comment highlighted in* blue *and the modified line, figure or table was also mentioned in each case.*
* * *
1. *General: However I believe the methods section needs to be improved for the work to be accepted.*

This section has been improved according to Q3 and further explanations has been added to insure clear methodology

2. *General: It would be very hard for others to reproduce this work reading the paper in it's current state.*

Unfortunately, we do not have a permission to share the airfoil geometry. It was very hard in the first place to get such data. We understand your concern, but we believe the methodology and the conclusions are beneficial to be shared with the scientific community. Moreover, the benchmark cylinder case is fully reproduceable. For the airfoil cases, we improved the description of each setup to help others to apply it on different cases in the future.

3. *Sections 2.1.1 and 2.1.2. : I don't think it's necessary for the authors to explain the turbulence models used: these are well known to readers experienced in CFD and can be researched by the interested reader in provided references if necessary. It would be preferable to briefly go over advantages and disadvantages of each model and to elaborate on why they were used in this specific case.*

We agree with the reviewer and this part has been reformulated, see new Sections 2.1.

4. Sections 2.1.1 and 2.1.2.: *Also, the choice of using two turbulence models in the same study seems odd, so further explanation on this aspect should be provided.*

Two different turbulence models were used to fit the different rwf's. Momentum rwf defines a relationship for turbulent kinematic viscosity, accordingly was used with SA turbulence model. On the other hand, DLR and Colebrook rwf's give relationship of k and omega at the wall, hence they were used with k-omega SST turbulence model. Also, it is very common to use SA in the IDDES hybrid models which seems to be promising for future work to use such a model with rwf. The main goal of this paper is to study the combination of widely used turbulence models in combination with rwf's to study the roughness effects. (Currently Sections 2.1)

5. Figure 2 : *The scanned data points shown in figure 2 for the two ice-profiles appear to be quite far apart. From what is stated in section 3.2 the scanned profile is used to calculate the average sand grain roughness. Please justify the expected impact of the number of scanned points on the modeling.*

The shown circles in this figure are just markers to differentiate the two profiles from each other. Since the real number of points is more than 1600 points for profile A and more than 2800 points for profile B, exact representation of each point was not shown. This figure has been updated with two different colors instead of markers to avoid any confusion.

6. Section 3.2: *How can one differentiate between roughness and geometry? In other words what part of the geometry can be justifiably smoothed and included in the roughness wall functions and what part must be retained? A robust criterion is not provided here. See final comments for further remarks on this.*

The smooth surface was generated using cubic spline. The cubic spline was selected in this case to make sure that the ice surface is smooth enough to avoid complex final shape and enable the usage of a large first wall cell. In case of a higher degree smoothing, the surface will keep some of its roughness and generating a mesh for such a surface will not result in a good quality cells near the walls. This methodology is comparable with the method used in the EN ISO 4287 standard for surface roughness measurements and calculations. The first cell height of each grid was indicated in Table 1. Also, this method has been highlighted further in the paper.

7. Section 3.2: *Assuming roughness elements with a conical shape seems quite semplicistic. Please justify this choice*

This assumption was based on scanned 3D ice surfaces shown in the article: "Convection from Surfaces with Real Laser-Scanned Ice Accretion Roughness and Different Thermal Conductivities" by Hawkins et al. that suggested either conical or hemispherical roughness elements and the article "Ice Roughness and Thickness Evolution on a Business Jet Airfoil" by McClain et al. In both articles, the scanned ice surfaces can be assumed to conical shapes to be able to calculate the parameters necessary to calculate equivalent sand roughness height. The two articles have been cited in the paper

8. Section 3: *Are rough wall functions used on the entire airfoil or just on the iced part? If it's the first, how does using rough wall functions on the entire airfoil influence the results? Perhaps it would be useful to check that all the wall functions proposed give reliable and consistent results for an airfoil with no icing.*

The rwf's were used only with the ice surface. The ice profile was separated as a different boundary and the rwf's were applied to them separately. If the rwf's are used with Ks = 0, the results should reduce to zero velocity shift. Also, the rest of the airfoil is smooth compared to the ice surface. Accordingly, any Ks value will be unrealistic. If the

roughness of the non-iced surface is present, the same wall functions could be definitely applied. It that case, it should represent the roughness of the coating or/and erosion.

9. Figures 4 and 5: *Please explain these figures better. If the "fine" and "coarse" grid refer to cases where wall functions are used vs cases where they are not the authors must explain the difference in the size of the elements surrounding the airfoil. The coarse grid differs not only for the boundary layer but also in the flow-field. In my opinion this can significantly influence the results.*

More explanation and grid data have been added to the next submission. (Currently Figures 5 and 6)

10. Figures 4 and 5: *The meshes appear different in the fine and coarse cases. "Fine" seems to use Cartesian cut-cell meshing, coarse uses polygonal cells. Please explain the differences.*

The fine mesh was used to simulate the flow around profiles in case of fully resolving roughness. This should give an indication about the benefits of using rwf's since using them will lead to minimal deviation from fully resolved roughness case while using a coarse computational grid. Currently (Figures 5 and 6)

11. Section 3.3: *Was a mesh independence study performed?*

Several grids with different first cell heights were studied in order to ensure the closest fit to Cl curve with the coarsest grid possible. However, the mesh test was carried out according to the criteria explained in the article "New Near-Wall Treatment for Suspended Sediment Transport Simulations with High-Reynolds Number (HRN) Turbulence Models" by Liu. These criteria have been further explained in the paper. The study is not mentioned since we have experimental data and fine mesh to give more confidence to our results Section 3.3

12. Table1 : *Why different Reynolds numbers?*

Since the roughness mainly effects on the detachment and re-attachment of flow on the surface, different Reynolds numbers should be a good idea to test the capabilities of each model. The different Reynolds numbers were provided by the experimental measurements.

13. Table1: *I suggest including the total number of elements of the mesh as well.*

It has been added to the revised version in Table1.

14. Figures 7,9 & 11: *The "Exp." Dataset appears to have duplicate data points for some AoA's. For instance, Fig. 9 AoA=12, 2 "Exp." Points are clearly visible. Please explain.*

The Experimental data includes two sets of measurements: one set in case of increasing and the second set for case of decreasing AoA. The big differences in Cl

values occurs only in post-stall AoA's due to high separation. The data has been plotted with error bars instead. (Currently Figures 8,10 & 12)

15. Figures 7,9 & 11: *Also, why are these figures relative to maximum lift? This could accentuate or diminish the difference between models depending on the value of maximum lift.*

We are not allowed to share the Cl values, but we reached an agreement with the experimental data owner to share the value normalized to Clmax. All values in each case are divided by a single value which is the Clmax of experimental measurements to maintain the correct relationship between the different cases. We have double checked the data to make sure that the relationship is correct. (Currently Figures 8,10 & 12)

16. Section 4.4: *As stated previously it is hard to judge differences in lift given the relative nature of figure 11. However, as it stands the differences seem pretty substantial even in the linear region.*

The Cl values is now recalculated taken into consideration only the summation of pressure forces since the experimental Cl values were calculated using pressure taps. A better agreement was shown in some case. However, some other cases had fair agreement due to the effects of the ice profile on the flow. A better understanding of these differences is shown in the Cp distribution results. Section 4.4

17. Figure 12: *here seems to be a considerable difference in figure 12b between momentum model and the others. This is not reflected in figure 11 at AoA=4°, why?*

Because in some cases, the deviation in Cp values on the upper and the lower surfaces can cancel each other and result in a summation of forces close to the experiments. Accordingly, the agreement analysis in Sec. 4.5 and Fig. 14 only use the Cp distribution as a criteria for agreement. A similar justification for these results was mentioned in line 273-277. (Currently Figure 13)

18. Line 240: *This phrase is not clear, please revise it.*

Noted and will be re-phrased. (Currently Line 247-250)

19. Section 5.1: *Authors state that many icing profiles should be measured and averaged to get a good reference shape of the airfoils. What about the roughness values? How can those be estimated and used in computations?*

The ice formation phenomenon is a stochastic phenomenon. This means that if the same airfoil was exposed to the same conditions many times, there should be small differences in the exact roughness shapes. Accordingly, to have an accurate simulation, many profiles should be considered and then average. However, this is not practically possible. Having said that, the estimated roughness should be less sensitive to such variation, since the estimation is kind of spatial average.

20. Line 260: *I suggest to edit figure 12 to include all of the tested roughness models. This would allow to compare if and where separation is predicted. Also, separation in this case seems like it is a consequence of the shape of the ice and not relative to roughness. If a model cannot correctly predict this separation can this be caused by excessive geometry smoothing? In other words, could agreement be improved is a different smoothing strategy was adopted? Or if no smoothing was adopted and roughness height measured differently?*

The separation bubbles are only visible in case of fully resolved flow around roughness because other rwf compensate the effects of these separation with a mathematical model to change the different turbulence parameters accordingly. Accordingly, there are only one figure can be generated with clear separation bubbles for each model at each AoA. Also, showing this figure in the article aims to given the reader some idea about the effect of irregular shapes on the surface. (Currently Line 268)

Line 263: *This is a good point. What kind of wall function is used in the non-iced part of the airfoils?*

for nu-t wall function: based on Spalding's law.

For k: wall function: zero-gradient with simple modifications (called kqRWallFunction in OpenFOAM v6).

For omega: based on Menter, F., & Esch, T. (2001). Elements of industrial heat transfer prediction. In Proceedings of the 16th Brazilian Congress of Mechanical Engineering (COBEM), November 2001. vol. 20, p. 117-127. (called omegaWallFunction in OpenFOAM v6).

---

## Author Comment (AC2)

**Responses to Reviewer Comments: Report #2**

*We would like to thank the reviewer for the effort in reviewing and commenting on this article. The comments provided have helped to improve the manuscript considerably.*

*Please find below our responses to each comment highlighted in* blue *and the modified line, figure or table was also mentioned in each case.*
* * *
1. General :*There are numerous grammatical / spelling errors in the text which must be corrected in addition to inconsistent notation.*

The article has been reviewed.

2. General: *The validation study applied was not necessarily particularly convincing, if possible it would be desired to see application to a more complicated geometry.*

In most of the literature, more simple geometries (like flat plate in a closed wind tunnel or airfoil with sandpaper roughness) was used which we thought it wouldn't be representative in this case. However, the validation cases used in this work were used in other literature like (da Silva et al. Cited in the paper). We thought that this experiment in particular should be close to our case since it represent an external flow case with relatively close roughness height values.

3. General : *The error metric chosen is in the view of the reviewer not particularly representative, as the relative error across the airfoil surface is not weighted based on location, which may lead to incorrectly displayed results and or results which are difficult to interpret.*

These plots have been updated using standard error equation to avoid high errors at experimental points with Cp close to zero. Also, the modified Fig. 14 takes into consideration only the points on the iced profile to give a closer look on the behavior of rwf's only

4. Line 18 :*This drop in Blades'*

 Corrected. Line 18

5. Line 19 : *Presence of rough ice surface*

Corrected. Line 19

6. Line 33 : *Form computational point of view*

Corrected. Line 33

7. Line 34 : *Ration*

Corrected. Line 34

8. Line 48 : *to no-slip condition*

Corrected. Line 48

9. Line 52 : *around iced wind turbine airfoil*

Corrected. Line 52

10. Line 74 : *Spalart*

This line is now deleted according to other reviewer's comment

11. Line 74 : *viscousity-like*

This line is now deleted according to other reviewer's comment

12. Fig. 1 : *Flow direction should be given in this diagram*

IT has been added in Fig. 1

13. Line 131 : *Equ.*

Corrected. Line 126

14. Fig. 3 : *Are local changes in height also accounted for within the modelling parameters? Or are deviations away from the mean line assumed to play a negligible role?*

The roughness deviations from the mean line are used to calculate average roughness height, diameter and pitch. These averages are then used to calculate equivalent sand roughness height which is fed to the roughness model. This has been elaborated on more clearly in Sec. 3.2. Fig. 4

15. Line 166 : *Paramedical*

Corrected. Line 174

16. Line 175 : *Between0*

Corrected. Line 183

17. Line 178: *This is not visible in the given plots and is very much a definition of how one would define a "good" agreement. I only really see the momentum model predicting somewhat representative $C_P$ values up to 70 degrees.*

Lines 175-179 have been reformulated. Line 183

18. Line 180 : *Spalart*

Noted and will be corrected. Line 188

*19.* Line 181 : *Model*

Corrected. Line 189

*20.* Line 182 : *that rough*

Corrected. Line 190

*21.* Line 208 : *over prediction: Word usage not consistent. Sometimes you write over prediction, sometimes underprediction (no space).*

Corrected. Line 208

*22.* Line 210 : *Figures 10a-10d: Again notation not consistent. Sometimes you write "Figures" sometimes fig.*

Corrected in the whole document.

*23.* Line 212 : *While the Momentum rwf :Consistency! Momentum rwf. Momentum RWF*

Corrected

*24.* Line 213 : *Formatting here not good. Huge empty page.*

Corrected. Line 220

*25.* Equ. 17 : *N is the measured points? Is this simply the number of points or is some type of integration scheme being used.*

Equ. 17 is the simple averaging equation and N is the number of of measured points. This will be indicated in the equation to avoid confusion. Equ. 14

*26.* Equ. 17 : *Furthermore, if this is over the entire airfoil is this really a suitable measure? The accuracy of the model is much more important near the leading edge, where d(Cp)/dX is the largest, and hence where the pressure distribution most greatly affects blade loading. The results, if weighted for regions of interest may be significantly different.*

Error chart (Fig. 13) has been replotted for the Cp errors in the iced region only. This figure has been updated and discussed in the next submission. The error criteria is shown in Equ. 14 and Sec. 4.5

*27.* Line 254 : *bad style (So, )*

Corrected, Line 264

*28.* Line 256 : *bubbles'*

Corrected. Line 266

*29.* Line 260 : *forms realtively*

Corrected. Line 272

*30.* Line 260 :*That's*

Noted and will be corrected. Line 273

*31.* Line 266 : *AoA's*

Corrected. Line 278

*32.* Line 270 : *The model overhead may be equivalent; the computational expense will however change eg. if a 1-2 equation model is being applied.*

Text has been updated. Line 281-283

---

## Author Comment (AC3)

[revised manuscript text omitted]
. In general aerodynamic simulation problems, both models show good results, especially in the pre-stall AoA's. However, each model has its own advantages. Since Spalart-Allmaras (SA) turbulence model is a one-equation model, it is usually more stable and requires less computational power. On the hand, k-$\omega$ SST can reach better results due to solving more turbulence parameters, but of course with more computational cost.

In this work, these two different turbulence models were specifically chosen to fit for different wall functions. As will be explained in the next section, one of the used RWFs defines a relationship for turbulent kinematic viscosity $\nu_t$. This type will be used only with SA model since it mainly solves $\nu_t$ equation to simulate the flow field. The other two RWFs adapts k and $\omega$ values near the wall for the presence of roughness. Accordingly, these two RWFs will be used only with k-$\omega$ SST turbulence model.

**2.2 Rough wall functions**

[revised manuscript text omitted]

**3.2 Roughness parameters calculations**

Since all RWFs treat the rough wall as smooth wall with a velocity shift as explained in Sec. 2.2, the actual rough surface of the ice profile is replaced with another equivalent smooth surface. The new smooth surface will be used to generate the computational grid around the profile and will be numerically treated as rough surface, i.e. a velocity shift will be added to the smoothened surface. To calculate this new surface, the rough surface was smoothened with cubic splines and . Similar to the calculation procedures of roughness parameters indicated in the DIN-EN-ISO 4287 standard (DIN (2010)), the rough surface should be smoothed as shown in Fig. 3a, using a cubic spline in this work, to find an average, smooth surface. After that, this smooth surface could be mapped to the x-axis as shown in Fig. 3b to ba able to analyze the roughness parameters.

Knowing the distance between roughness elements and height of elements, average sand roughness height explained in Fig.1 can be calculated using Eq. (1). By analyzing the laser scanned ice surfaces in different literature like Hawkins et al. (2017) and McClain et al. (2018), it can be assumed that roughness elements take conical shapes. Accordingly, $A_p = \pi D_{avg}^2/4$ and $A_s = 0.5 K_{avg} D_{avg}$. The above analysis gives results in $K_s = 1 \times 10^{-3} m$ and $1 \times 10^{-2} m$ for profiles 1 and 2 respectively.

[Figure]

(a)  (b)

**Figure 3.** An example of: (a) smoothing a rough surface using cubic spline and (b) mapping roughness to x-axis

**3.3 Grid generation**

In order to use rough wall functions, the height of the first cell center should be large enough to cover the roughness element. Along with converting the rough surface into a smoother one, the resulting grid is much coarser than the grid required by smooth wall functions which require $y^+(1) < 1$ to be able to correctly simulate the boundary layer. Accordingly, the studied approach in this work requires less computational resources. In case of the two ice profiles studied in this work, to properly generate a grid that fulfills the condition of $y^+(1) < 1$, is found to require number of cells $\approx 4 \times 10^5$ cells in 2D simulations. On the other hand, when using rough wall functions with the proper first cell height and roughness smoothing, it is found to

[Figure]

[Figure]

(a)     (b)

**Figure 4.** Comparison between original and smoothened ice surfaces for: (a) profile 1 and (b) profile 2

155 require less than $1 \times 10^5$ cells in 2D simulations. A comparison between smooth and rough wall function grids is shown in figures 4 and 5. Resulting from the numerical setup described above, properties of computational grids used in this work are given in Tab.1.

To provide a comparison between the results of fully resolved surface roughness and using RWFs, the next section will also provide a comparison between $C_l$ values resulted from using fine and coarse grids, shown in figures 5a and 6a respectively,

160 and coarse grids, shown in figures 5b and 6b, used with RWFs.

It should be noticed that the first cell height in case of RWF simulations is relatively larger that in case of fully resolved mesh to fit the criteria explained by Liu (2014). This criteria mandates that the height of the cell center of the first cell near the wall should be larger than the roughness height. Accordingly, a mesh independence test was done to select the first cell height values indicated in Tab 1.

165

**Table 1.** Simulation cases and its grids

| Case No. | Profile | Re | AoA [$^o$] | y(1) [m] | | No. of Cells | |
| --- | --- | --- | --- | --- | --- | --- | --- |
| | | | | Resolved | RWF | Resolved | RWF |
| 1 | profile 1 | $2.6 \times 10^6$ | -4$^o$ to 16$^o$ | $1.57 \times 10^{-5}$ | $9 \times 10^{-3}$ | $4.1 \times 10^5$ | $7.9 \times 10^4$ |
| 2 | profile 1 | $5.2 \times 10^6$ | 0$^o$ to 16$^o$ | $1.22 \times 10^{-6}$ | $9 \times 10^{-3}$ | $4.6 \times 10^5$ | $7.9 \times 10^4$ |
| 3 | profile 2 | $3.1 \times 10^6$ | -4$^o$ to 16$^o$ | $1.56 \times 10^{-5}$ | $10 \times 10^{-3}$ | $4.1 \times 10^5$ | $6.3 \times 10^4$ |

[Figure]

(a)                (b)

**Figure 5.** Comparison between: (a) fine (with fully resolved roughness) and (b) coarse (with RWFs) grids of profile 1

[Figure]

(a)                (b)

**Figure 6.** Comparison between: (a) fine (with fully resolved roughness) and (b) coarse (with RWFs) grids of profile 2

[revised manuscript text omitted]

As shown in the Figures 14a and 14b, it can be noticed that the higher the AoA value, the more error between simulation and

wind tunnel results except for some results at AoA = $4^o$ in case 1. While in Fig. 14a, we can see that the error values are not increasing with the increase of AoA in the same rates as shown in the previous two figures. These error results shows that all RWFs has a limited capabilities in simulating the detachment of the flow and accordingly give results deviated from actual results. Also, in case 3, where profile 2 was simulated, the flow should be highly unsteady which is not suitable for RANS
250   simulations. However, the Momentum RWF managed to predict the $C_p$ distribution better than the other models except for AoA = $4^o$.

[Figure]

**Figure 14.** Average error between experimental and simulation results of $C_p$ distribution (a) case 1, (b) case 2, and (c) case 3

**5   Discussion and conclusions:**

In this work, three different rough wall functions were tested on iced wind turbine airfoils. The aim of this comparison was to find out which RWF will be the most suitable one to simulate ice accretion on wind turbine blades exposed to icing atmospheric
255   conditions. To be able to apply these RWFs, DLR and Colebrook RWFs were implemented to OpenFOAM v6 CFD framework along with the existing Momentum RWF. After that, two ice profiles that collected from wind site, molded to airfoil, and

tested in the wind tunnel were smoothed by a cubic spline to find the equivalent smooth surface. Also, roughness parameters described in Sec. 3.2 were calculated and used to calculate the velocity shift value $\Delta u$.

In this section, general remarks and discussions of the results are introduced. Then the conclusions from the outcomes of this work will be highlighted.

**5.1 General remarks and discussion**

Regarding the rough ice surface shown in Fig. 2, one should keep in mind that the ice formation process (or roughness formation in general) is a stochastic phenomenon. Which means, if the same airfoil exposed to the same atmospheric conditions, the ice profile resulting will not be exactly the same. Hence, to find the real smooth surface required for the simulations, a large number of ice profiles of the same atmospheric conditions should be studied and averaged to find the real average surface. Also, each rough surface will give different attachment and reattachment bubbles locations, and might have different overall pressure distribution. However, the scope of this work is only to prove that the RWF approach results in good results compared to experiments.

Fig. 15 shows the LIC of flow in cases 2 and 3 at AoA = $0^o$ in fully resolved grid case only. In case of coarse grids, such separation bubbles will not be clearly visible since the RWFs compensate these effects with a mathematical model affecting on the different turbulence parameters. From the geometry of the two ice profiles shown in Fig.2, it can be noticed that both profiles are slightly inclined downwards. This inclination forms a relatively big separation bubble behind the profile on the lower surface as shown in Fig. 15. That is why we can see deviations between simulations and experimental results of $C_p$ distribution in this region (i.e. the lower surface between x/c = -0.1 to 0.1) in most of the studied cases.

It can be noticed from the results that agreement of $C_l$ curves between simulations and experimental results doesn't necessarily mean that flow field was accurately simulated. For example, in case 3, DLR RWF shows better $C_l$ agreement than other models at AoA = $4^o$ while from $C_p$ distribution, DLR RWF showed larger deviations from experimental results. The same happened with case 2 at AoA = $0^o$ and $4^o$. The overall $C_l$ at these AoAs in the end had good agreement because the deviations on the upper and lower surfaces compensate each other which can be misleading in this case. To accurately access the $C_l$, it should be studied together with the $C_p$ distribution curves.

It is noteworthy that all the RWFs used are only algebraic equations that express the behavior of the flow near that wall. Accordingly, the three different RWFs have the same computational cost. However, the overall computational cost depends only on number of cells in the used computational grid and the turbulence model itself as discussed in Sec. 2.

**5.2 Conclusions**

The simulation cases shows that DLR RWF provides the best agreement of cases 1 and 2 simulations with experiments compared to the other models. While in case 3, Momentum RWF provides the least errors between experiments and simulations.In some cases, the simulations show fair agreement at locations that witness violent separation like concave areas between ice and airfoil. This fair agreement is expected due to using steady state RANS simulations.

[revised manuscript text omitted]